# A Tactile Device Generating Repulsive Forces of Various Human Tissues Fabricated from Magnetic-Responsive Fluid in Porous Polyurethane

**DOI:** 10.3390/ma13051062

**Published:** 2020-02-27

**Authors:** Yu-Jin Park, Ji-Young Yoon, Byung-Hyuk Kang, Gi-Woo Kim, Seung-Bok Choi

**Affiliations:** Smart Structure and Systems Laboratory, Department of Mechanical Engineering, Inha University, Incheon 22212, Korea; pyjn5059@naver.com (Y.-J.P.); ji_young62@naver.com (J.-Y.Y.); 1357op@gmail.com (B.-H.K.)

**Keywords:** tactile device, magnetorheological fluid (MRF), porous polyurethane foam (PPF), squeeze mode, repulsive force, human tissue, psychophysical test

## Abstract

In this study, a controllable tactile device capable of realizing repulsive forces from soft human tissues was proposed, and its effectiveness was verified through experimental tests. The device was fabricated using both porous polyurethane foam (PPF) and smart magnetorheological fluid (MRF). As a first step, the microstructural behavior of MRF particle chains that depended on the magnetic field was examined via scanning electron microscopy (SEM). The test samples were then fabricated after analyzing the magnetic field distribution, which was crucial for the formation of the particle chains under the squeeze mode operation. In the fabrication of the samples, MRF was immersed into the porous polyurethane foam and encapsulated by adhesive tape to avoid leakage. To verify the effectiveness of the proposed tactile device for appropriate stiffness of soft human tissues such as liver, the repulsive force and relaxation stress were measured and discussed as a function of the magnetic field intensity. In addition, the effectiveness and practical applicability of the proposed tactile device have been validated through the psychophysical test.

## 1. Introduction

In recent years, operations using advanced tools and robots have been gradually increasing in the medical industry. Among them, the laparoscopic and robotic surgery, which are considered the revolution of surgical methods, are becoming popular [1,2]. Abdominal operations are normally performed by opening a large incision in the abdomen [3]. However, with the advancement of surgery technologies, a small incision of 0.5–1.5 cm can be used to insert a camera and other tools into the abdomen [1]. The range of vision can be enlarged up to 15 times during surgery [2,4]. Therefore, even if the surgical site is small, its state is properly grasped, and surgery proceeds without difficulty [1,3]. Moreover, the hand movements of the surgeon performing the surgery are digitized, and thus shaking can be prevented [1]. This type of surgery is also called minimally invasive surgery (MIS) because it is performed with minimal incisions. For instance, the laparoscopic and robotic surgeries have a smaller incision size compared to the laparotomy, and the surgical site’s pain is much less [3,5,6]. In abdominal surgery, the surgical site is enlarged to make the surgeon’s vision clearer. In addition, if the surgery site is deep, organs covering the surgical site might need to be removed, which can hinder or slow down the patient’s recovery [1,3,6]. With the introduction of MIS, the surgeon is able to operate with minimal incision, and thus the patient can recover faster compared to abdominal surgery. In conventional MIS, the visual field seen in the monitor is a two-dimensional (2D) image, and thus it is difficult to know the distance between structures [7]. The surgeon should intuitively judge the distance of surrounding structures to pinpoint the surgical site [4]. In abdominal surgery, surgeons directly touch the surgical site and see it in front of them, which allows immediate identification and response [4]. However, in MIS, it is difficult to determine exact state of the surgical site because all information is judged only by the enlarged visual element. To compensate for this, three-dimensional (3D) imaging provides real-time information. It uses multiple cameras in various angles [8,9,10,11], thus increasing the quality of information delivered to surgeons. However, this technique still communicates only by visual information. In the face of the limitations of visual information transfer, tactile sensing and haptic devices are required. If remote real-time control of a tactile device is possible, a tactile device can be developed based on the information of the surgical site during laparoscopic surgery or robotic surgery. All sensed information is visually presented to the surgeon. The repulsive force received from the surgical tool in the surgical site is approximately 0–5 N [12,13,14]. It is known that human tissues have viscoelastic properties and stress relaxation is occurred against external stimuli [15,16,17,18,19]. If a device can realize a repulsive force within this range and a stress relaxation similar to human’s tissue, this device could be directly applied to the haptic master part of the MIS or the laparoscopic surgery to transfer the state of the tissue to the surgeon hand’s sense [20,21].

Various methods have been proposed for devices to realize tactile sensations [20,21,22,23,24]. The most widely researched one is the shape memory alloy (SMA) wire technology [20,22,24]. However, this method has the disadvantages of low bandwidth and poor performance when using a mechanical amplification system to increase the required stroke [22]. The second most widely studied method uses piezoelectric materials. This method can present a wide range of force and enough force, but it does not realize the relaxation force. Moreover, techniques using micro-electro-mechanical systems (MEMS), electro-rheological fluids (ERF), or polymers are being studied, but have not been widely investigated or practically used [22,25]. Despite the high number of developed devices, they are not suitable for surgical robots or laparoscopes because they lack viscoelastic properties to transfer the information from the patient’s surgical site. Recently, a pneumatic haptic feedback system has been proposed [21]. In that work, the tactile was controlled by using an actuator array of a balloon face and adjusting air pressure. More recently, a tactile device utilizing magnetorheological fluid (MRF) has been proposed for application in robot-assisted MIS (RMIS) [26]. The use of MRF in these proposed tactile devices has several advantages. For instance, viscoelastic characteristics of MRF can be quickly controlled by the magnetic field with wide range of change and low power consumption [27]. Therefore, MRF-based tactile devices can represent various organ characteristics of human tissues, and its controllability can realize certain stiffness and damping properties in a simple open-loop control method. Despite the advantages of the proposed MRF based tactile device, the proof-of-concept investigation was only developed so far without considering the possible practical availability in RMIS. Figure 1 shows a schematic flow of the RMIS with the proposed tactile device. It is seen from the figure that the repulsive force of the human organ tissue that occurs during the operation needs to be felt by the surgeon. Currently, there is no such function in a robotic surgery system.

The main contribution of this work is the fabrication and testing of a controllable tactile device which can capture several different repulsive forces of human tissues that occur at the operating site in RMIS or any other robotic-assisted surgery. To achieve this goal, tactile samples were fabricated by immersing MRF into porous polyurethane foam (MRP, in short), and then encapsulating it by adhesive tape to avoid leakage. Prior to fabricating the samples, the magnetic field distribution of MRP was analyzed using a proper magnetic core circuit to ensure uniform field distribution. Subsequently, an experimental apparatus to test the compressive force of MRP samples, whose stroke in the vertical direction is controllable, was manufactured by attaching the sensors of the load cell and the linear variable differential transformer (LVDT). The field-dependent compressive force of the samples was then measured as a function of current magnitude (or magnetic field intensity). After analyzing the controllable force spectrum of the samples, their field-dependent compressive forces were compared with soft tissues of human organs such as the heart. In order to investigate the repulsive force spectrum of the proposed MRP device, nine different human tissues, such as liver and heart, were adopted as comparative objects [28,29,30]. In other words, the maximum repulsive forces generated from the proposed MRP device at different magnetic fields were compared with Young’s modulus (equivalently maximum repulsive force) of nine human tissues to demonstrate the practical applicability of the proposed tactile device. In addition, to verify the practical feasibility, the psychophysical tests were performed, and the results were discussed. It is noted here that even though the surgeon is able to use the proposed tactile device for the minimally invasive surgery, the test to recognize the different repulsive force depending on the magnetic field intensity (or current input) needs to be carried out to identify the corresponding repulsive force of the human organs or tissues.

## 2. Fabrication of MRP Samples

### 2.1. Operating Principle

In general, there are three different operation modes in application devices or systems utilizing MRF. They are classified into flow, shear, and compressive (or squeeze) mode. They can be determined by the direction of the force acting between the two magnetic cores. For example, the magnetic core is fixed for the flow mode, but it is moved in the horizontal direction for the shear mode. In the squeeze mode, the upper core is vertically moved while the low core is fixed. Therefore, each mode has an inherent property and thus can be applied to different devices or systems. The flow mode can be applied to damper and shock absorber because it controls the channel to which the magnetic field is applied. The shear mode is useful for brakes and clutches because one of the cores rotates in these devices. The squeeze mode is suitable for controlling large forces on the vertical axis, although the controllable displacement is small. In fact, different tactile devices can be fabricated using the different operation modes of MRF. Among the three modes, the squeeze mode is the most suitable for the tactile device proposed in this work because the surgeon can feel different repulsive forces by simply compressing the device with different magnetic fields. In other words, if the surgeon’s hand exerts a force in the same axis as the direction of the magnetic field of the tactile device (squeeze mode), it is possible to reproduce different repulsive forces as a function of the magnetic field (or input current). Such a squeeze mode based tactile device could mimic the viscoelastic characteristics of the internal organs of a human body and reproduce accurate tactile information. Therefore, this study proposes a tactile realization device using the squeeze mode operation of MRF. Figure 2 presents a simple schematic configuration of the proposed MRP tactile device. It is assumed that the hand of the tester is located in the same axis as the direction of the magnetic field. The MRP is located where the core portion of the magnetic field vertically exits the electromagnet. The electromagnets are chosen by considering both the required magnetic field and the size of the MRP.

### 2.2. Fabrication Procedures

Figure 3 shows a simple procedure for the fabrication of the MRP sample. Biochemical filter polyurethane forms of 25 and 100 ppi (25 and 100 pores/${\mathrm{inch}}^{2}$) were used as the default framework. The polyurethane foam was wrapped with an adhesive tape (LDPE: Low Density-Polyethylene, Cleanwrap Co., Kimhae, Korea). This allowed MRF to be retained in the pore when MRF was injected into the open cell type foam. In addition, this procedure helps restore the original shape when a certain strain is removed. Subsequently, MRF was injected using a syringe (Kovax-Syringe 10 ml, Korea Vaccine Co., Ltd., Seoul, Korea). The injected parts were sealed by the polypropylene self-adhesive tape (Dongyang Tape Co., Sejong, Korea), and the fabrication was then complete. By placing the sealed portion in the part in contact with the electromagnet, the concerns regarding surface resistance due to sealing were eliminated. The size of the completed MRP was 2.5 cm × 2.5 cm × 1 cm, which was determined by considering the area of the magnetic field from the electromagnet used in the experiment. The MRF used in this work is of commercial fluids, model MRF-122EG, Load Corporation. In this fabrication, several different types of polyurethane foams were tested to adjust the appropriate initial force. It is noted here that the proposed tactile device can be made from the size of $1\;cm^{3}\;$ to $16\;cm^{3}$ to produce the repulsive force of the human tissues. An appropriate size can be determined by the surgeon’s preference to touch and feel the stiffness of the human tissues.

### 2.3. Microscopic Observation

To understand the reasons for the control of the repulsive force by the magnetic field, the MRF and polyurethane foam were microscopically observed. Figure 4 shows scanning electron microscope (SEM, S-4300SE, Hitachi High-Tech Co., Tokyo, Japan) images of the particles in MRF. To obtain the SEM images, a simple frame with powerful magnets was produced, in which a small specimen was placed in the transverse direction. In addition, MRF was diluted for a clear vision of the particles. It is clearly seen from figures that the particles are well distributed in the carrier liquid in the absence of the magnetic field. The particle size ranged from 10 to 50 µm. But, as soon as the magnetic field is on, the formation of the chain-like structures between the particles according to the direction of the magnetic field occurred. The magnetic field was applied in the transverse direction, and the chains were formed in the same transverse direction as the magnetic field. The chain is a collection of particles of various sizes, which can be seen as a dense structure along the path of the magnetic field. When the chain is formed, the behavior of MRF changes from Newtonian to Bingham fluid, which presents field-dependent yield stress. Therefore, the rheological properties of MRF such as storage modulus can be controlled by the intensity of the magnetic field. In the proposed MRP tactile device, the compressive force depends on the applied magnetic field intensity and, consequently, various repulsive forces of human tissues can be generated.

Polyurethane foam was used to maintain the morphology of the MRP and mimic the morphology of human tissues. The polyurethane foam is a structure in which many pores of irregular size are entangled. SEM images of the polyurethane foam were obtained to confirm the shape of the pores. Figure 5 presents SEM images of two different polyurethane foams: one of 25 ppi and the other of 100 ppi. There are many pores in the structure where MRF is to be contained for the proposed tactile device. In addition, the stem constituting the pores of 100 ppi is approximately five times thinner than that of the polyurethane foam of 25 ppi. Therefore, the polyurethane foam with 100 ppi presented a higher initial force than the one with 25 ppi. In this study, two samples of polyurethane foam based on different ppi were fabricated and tested to investigate the initial force and force controllable range. It was noted that uneven pores were filled with MRF, and the shape of the pores could be restored over time.

### 2.4. Magnetic Analysis

The electromagnet used in this work can produce a magnetic density of up to 0.2 T. (JL-10A, DC90, JL. Magnet Co., Seoul, Korea). The magnetic field generated from the electromagnet should be uniformly formed in the MRP. To confirm the uniform distribution of the magnetic field, a magnetic model based on the input current was created, and a magnetic field analysis was conducted. For that, a copper wire of 0.7 mm diameter was wound approximately 2250 times. In this case, the proposed model could produce a magnetic density of approximately 0.2 T for an input current of 0.4 A. The analysis results are shown in Figure 6. The magnetic density was vertically positioned in the MRP, and its maximum value was identified at approximately 0.2 T. Despite the field being formed radially toward some corners, the center is almost perfectly vertical. Therefore, the direction of the magnetic field between the two plates is in the same axis in which the force is applied, thus representing the basic type of the squeeze mode.

### 2.5. Fabricated Samples

To analyze if the MRP samples were well manufactured, the field-dependent particles chains of MRF filled in pores need to be observed. In other words, when the magnetic field is formed in the MRP, it is necessary to check the particles behavior of MRF filled in the pores. Figure 7a presents an MRP sample placed on the electromagnet. The hemisected sections of the MRP sample are presented in Figure 7b,c without and with the magnetic field, respectively. It can be confirmed that when the magnetic field is off, MRF is absorbed inside the pores of the polyurethane foam. In contrast, when the magnetic field is on, the particles of MRF form chain-like structures in the direction of the magnetic field. It was clearly and visually confirmed that the forming direction of the MRF particle chains and the direction of the applied force are the same. This behavior of particle chains is similar to the magnetic field analysis results shown in Figure 6.

## 3. Repulsive Force Measurements

### 3.1. Experimental Apparatus

To measure the characteristics of soft tissues from human organs, stress relaxation is frequently adopted because, despite being a simple method, it represents the repulsive force characteristics of human tissues well. Figure 8 presents the experimental apparatus to measure the field-dependent repulsive forces of the proposed MRP tactile device. As shown, the LVDT sensor (MTA-5E-5KC-MB, Celesco Transducer Products, Inc., Toronto, Canada) is used to measure the distance in the compressive direction, and the S-Beam type force sensor (LSB205, 0N-22.2N, Futek Advanced Sensor Technology Inc., California, USA) is used to measure the repulsive force. The power supply for the force sensor and LVDT is used as the input source. The programmable power supply for the input current and voltage of the motorized test stand and for the electromagnet (NF. EC750SA, Programmable AC/DC Power Source, NF Corporation, Inc., Yokohama, Japan) is used with the data collector to send the signal, and the motor is controlled by using a microprocessor (ACE kit 1104 CLP, dSPACE GmbH, Inc., Paderborn, Germany). The experimental conditions were as follows. The MRP samples went separately through a compressive strain of 1 mm. The stress relation was measured for 120 s to observe compression deformation. The endpoint touching the top surface of the MRP moved up and down in the vertical direction at 0.4 mm/s. The magnetic field range applied to the MRP sample ranged from 0 to 0.2 T.

### 3.2. Results and Discussion

Figure 9 shows the controllable range of the repulsive force achieved from MRP samples of 25 (MRP1) and 100 ppi polyurethane foam (MRP2). To achieve this result, an input current of 0–0.4 A was applied under 1 mm deformation. The results showed that the peak force values ranged from 0.25 to 0.70 N for the MRP1 sample, and from 0.20 to 1.06 N for MRP2. The stress relaxation value of MRP1 ranged from 0.14 to 0.35 N, and that of MRP2 ranged from 0.11 to 0.65 N. In addition, it can be clearly observed that all relaxation stresses were stably formed within 120 s. This allowed soft human tissues to be mimicked within the controllable range of MRP1 and MRP2. Moreover, MRP2 showed lower initial value and higher repulsive force because its MRF was located between smaller pore stems. As a result, the particle chains of MRF were more firmly formed when the magnetic field was applied to the fluid domain. Consequently, MRP2 presented a wider controllable range of the repulsive force spectrum than MRP1 for the same input current.

Figure 10 shows the field-dependent stress relaxation results of the MRP2 sample under 1 mm deformation. The stress relaxation was formed within 120 s for an input current of 0–0.4 A. As expected, the instantaneous repulsive force was the highest at 0.4 A. Moreover, the higher the magnetic field applied to MRP2, the longer the time for the relaxation formation. This phenomenon occurred because the stronger the elastic characteristic, the longer the stable stress relaxation formed at a given constant strain. Therefore, upon application of a high magnetic field (or input current), the material characteristic of MRP2 were expected to become tough. The peak and stress relaxation values of the resulting graph in Figure 10 were identified and summarized in Table 1. The maximum peak force ranges from 0.2 to 1.06 N, and the stress relaxation from 0.11 to 0.65 N, in 120 s. In all conditions, stable stress relaxation was formed within 120 s. Furthermore, the stress relaxation values were formed at 55–61% of the position of the peak force value. It is noted that the peak value was determined by adopting Max function and the stress relation value was achieved as follows. Since the repulsive force is converged to at 120 s as seen from the measured results, the stress relation value is obtained as an average value of two values at 110 s and 120 s.

### 3.3. Validation of Practical Use

Based on reference data [28,29,30], Young’s moduli of nine different human tissues are given in Table 2. A finite element method analysis was conducted to convert Young’s moduli into the repulsive force corresponding to the measured values in this work. The model used for the finite element method analysis is of the same size as the actual experimental apparatus. Figure 11a shows the end effector compressing MRP by 1 mm. The diameter of the end effector is 14 mm, which was determined based on the thickness of the index finger of a human. Figure 11b shows the mesh elements of the area where the end effector is forced to reach the surface of the MRP and compress it. The total number of nodes related to this area is 623, and the mesh element size is created at 1 mm. The repulsive force applied to this area can be obtained from the following equation. Table 2 shows the calculated maximum repulsive force for each tissue obtained through the finite element analysis. These values can be compared with the maximum peak force value measured in this work. Additionally, the magnetic flux density applied to MRP2 corresponding to the calculated maximum repulsive force of human tissues is also shown in Table 2.
(1)$$F_{Repulsive\;Force}=\sigma_{Average\;of\;Elements\;value}\times A_{End\;Effector}$$

Figure 12 shows the schematic comparison of the repulsive forces that occur in the human tissues according to the measured results and calculated values from the finite element analysis. The experimental results are focused on the controllable ranges of MRP1 and MRP2. The controllable range of MRP1 was within 0.25–0.70 N, and it included some parts of kidney, skin, skeleton muscle, and heart. MRP2 presented a wider range than MRP1, at 0.20–1.06 NR. MRP2 covers smooth muscle, kidney, skin, skeleton muscle, heart, and intestine. The proposed MRPs present improved performance by widening the controllable range. Research was conducted to mimic the tactile of as many types of tissue as possible with MRP2. However, the very soft tissues, namely lung, fat, and brain, still presented high initial values and could not be realized. Nevertheless, smooth muscle, kidney, skin, heart, skeletal muscle, and intestine were mostly included in the force range of MRP2. After analyzing the controllable force spectrum of the sample, its field-dependent compressive force was compared with the soft tissues of human organs to validate the proposed tactile device for robotic surgery. Research was conducted to low initial value to broaden the force range. However, some tissues, such as lung, which have very low repulsive forces, could not be realized from the proposed tactile device. In order to resolve this issue, the following problems need to be further explored. First, the current polyurethane mesh is too rigid to express the stiffness of soft tissues. Thinner stems and smaller pores than the currently used ones should be considered. However, they have the following limitations: If MRP is manufactured with a softer foam with low initial value, the frame might not be supported, and adverse effects may be generated. Therefore, finding a suitable foam is the most important task to address this issue. Secondly, the film that seals the polyurethane is too elastic. Regardless of the foam softness and the low density of MRF, it may be difficult to reduce the initial value if the surrounding material is too elastic. Third, the sealing adhesive tape affects the stiffness of MRP samples. This tape may have a higher initial value because the surface is not elastic. Consequently, the above problems should be further investigated to have a wider spectrum of the repulsive force of the proposed tactile device.

### 3.4. Psychophysical Test

It is known that psychophysical tests are frequently adopted to verify the effectiveness of the tactile devices and haptic systems. In this work, MRP2 sample was used for the psychophysical test to identify the repulsive forces of different human tissues; muscle (0.22N), heart (0.37N), and intestine (0.68N). The test was conducted with 20 volunteers between 20 and 30 years old by dividing two groups based on the training time; group A (15 min training), and group B (30 min training). The two different training times of 15 min and 30 min are relatively short compared with other tactile devices. The reason why such as short training time is adopted in this work is to show how fast normal people (not surgeon) can identify the field-dependent repulsive force (or stiffness of the tactile device) through a blind test. After training, all volunteers rested for 2 h and the blind test was carried out for the recognition of the tissue he or she touched. The standard deviation was obtained by randomly implementing the repulsive force of the three organs and recording the average of the number of recognized. The results are shown in Figure 13. The longer the training time, the better recognition, as expected. After the testing, the specific questions presented in Table 3 were made. This question was recorded on a five-point scale (very poor: (1); poor: (2); good: (3); very good: (4); excellent: (5)), with the mean score for the response and the standard deviation. The questions at this stage are: Question number 1 is “Can you recognize the difference between repulsive forces?”; and question number 2 is “Do you agree that the repulsive force for each organ can be realized using the proposed sample?”. 

The results presented in Figure 13 and Table 3 are quite self-explanatory, justifying that the different repulsive forces depending upon the different input current can be easily and accurately recognized from the proposed tactile device. In addition, an accurate recognition of the repulsive force from the sample without long training time is one of the advantages, and thus the proposed tactile device can provide high practical feasibility in the future. It is here noted that since normal people can accurately distinguish the different stiffness from the proposed tactile sample, the surgeon can easily identify the stiffness of the human tissues with a very short training session. This advantage can bring the reduction of surgery time with high accuracy in the robot-assisted minimally invasive surgery.

## 4. Conclusions

In this work, a new type of tactile device using MRP with the squeeze mode was proposed, and its practical applicability was validated by comparing the measured repulsive force with those of nine different human tissues. After fabricating two MRP samples with polyurethane foams of 25 ppi or 100 ppi, SEM images of MRF and polyurethane foam were analyzed to understand the controllable repulsive forces depending on the intensity of the magnetic field. Subsequently, an experimental apparatus was established to measure the field-dependent repulsive force, and the stress relation test was performed. From the measured stress relaxation results, the field-dependent repulsive force and maximum peak values were obtained. The tests showed that the controllable range of the maximum peak was ranged from 0.25 to 0.7 N for MRP1 at 25 ppi, and from 0.2 to 1.06 N for MRP2 at 100 ppi. Therefore, the MRP2 sample can cover wider spectrum of repulsive forces of soft tissues. In other words, MRP2 has a wider controllable range of repulsive force under the same magnetic field intensity. From the schematic comparison chart between the measured values of the proposed tactile device and the calculated values of human tissues, it has been identified that MRP1 is more suitable to mimic very soft tissues such as lung, whereas MRP2 is suitable to mimic most human tissues, including skin and heart. In order to demonstrate the practical feasibility of the proposed tactile device, the psychophysical test was also carried out using MRP2 sample. It has been shown from this test that one can accurately recognize different repulsive forces after training a short period; 15 min and 30 min. This result is very promising for practical applicability of the proposed tactile device to the robot-assisted MIS and other MIS operation. It is finally remarked that the analytical model of the proposed MRP tactile device will be explored in the near future as a second phase of this work.

## Figures and Tables

**Figure 1 materials-13-01062-f001:**
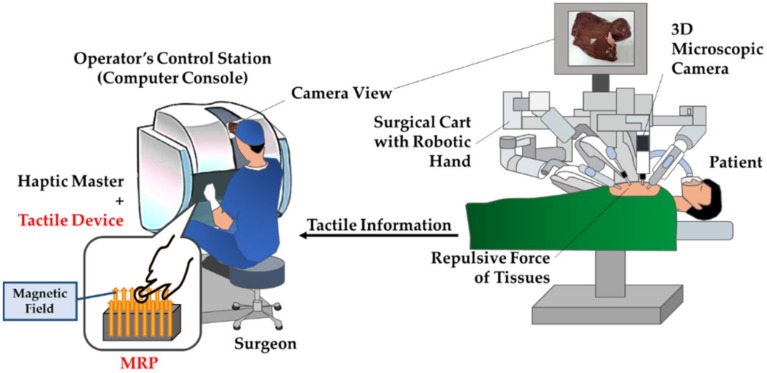
Robot-assisted minimally invasive surgery (RMIS) schematics with the proposed MRP tactile device.

**Figure 2 materials-13-01062-f002:**
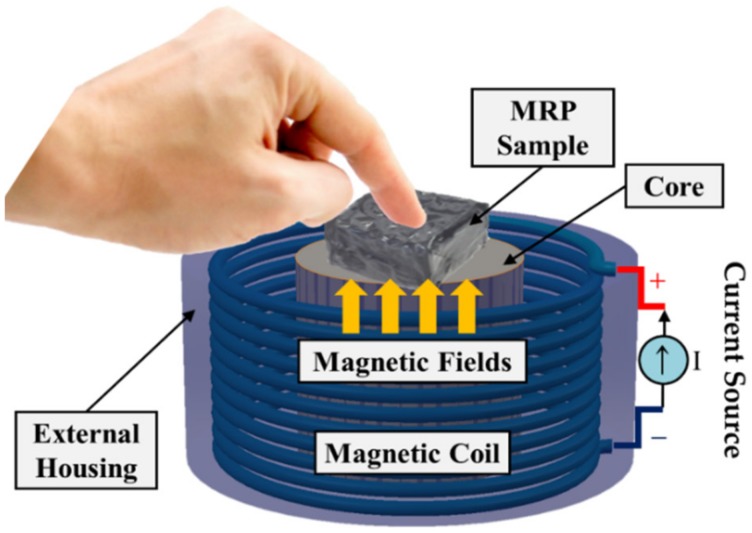
MRP sample subjected to the magnetic field

**Figure 3 materials-13-01062-f003:**
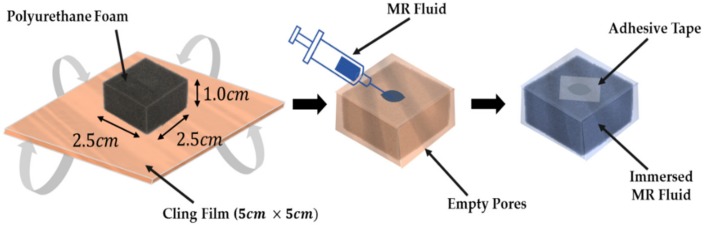
Procedures of MRP fabrication.

**Figure 4 materials-13-01062-f004:**
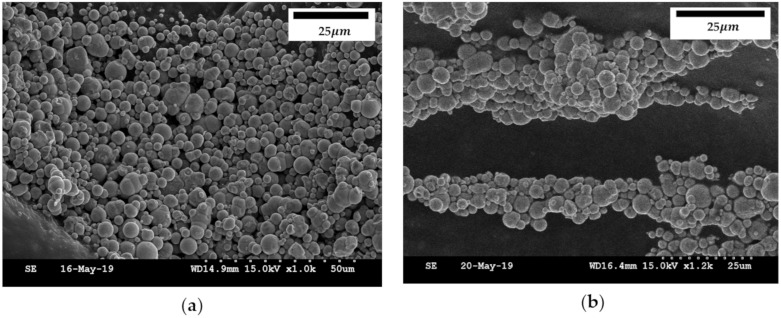
SEM images of the particles in MR fluid; (**a**) magnetic field OFF, (**b**) magnetic field ON: 0.1 T.

**Figure 5 materials-13-01062-f005:**
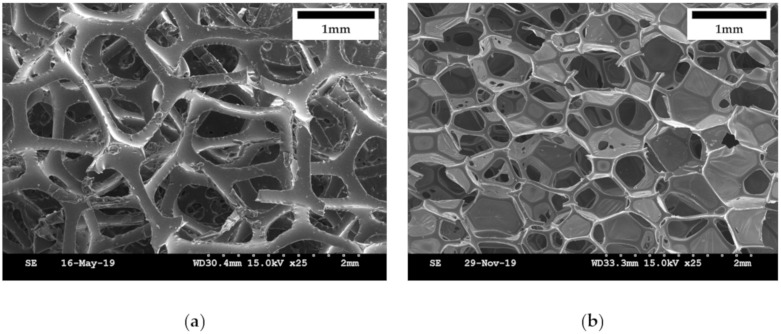
SEM images of polyurethane foam: (**a**) 25 ppi, (**b**) 100 ppi.

**Figure 6 materials-13-01062-f006:**
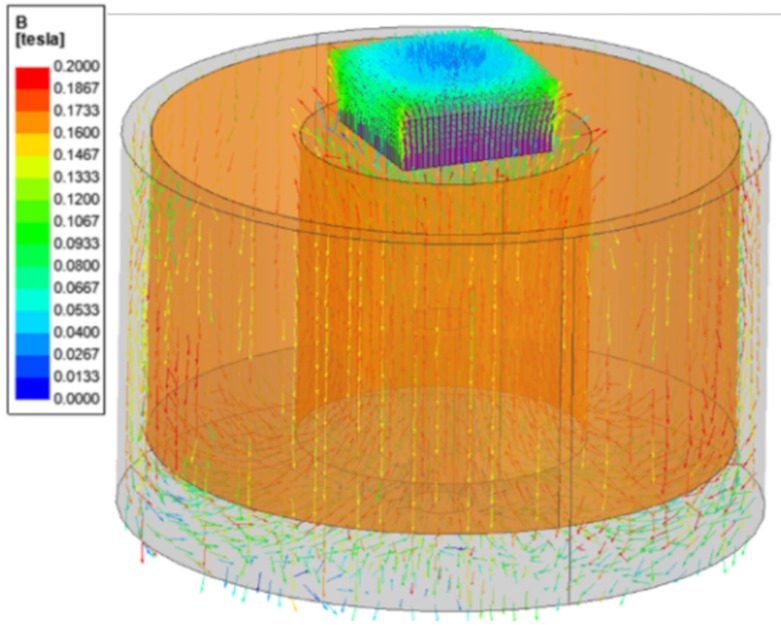
Magnetic field analysis of MRP tactile device.

**Figure 7 materials-13-01062-f007:**
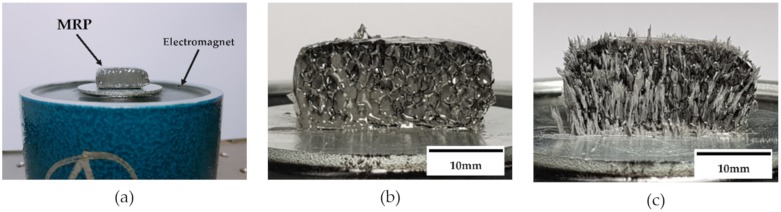
Fabricated MRP sample: (**a**) Sample with electromagnet, (**b**) section view at magnetic OFF, (**c**) section view at magnetic ON.

**Figure 8 materials-13-01062-f008:**
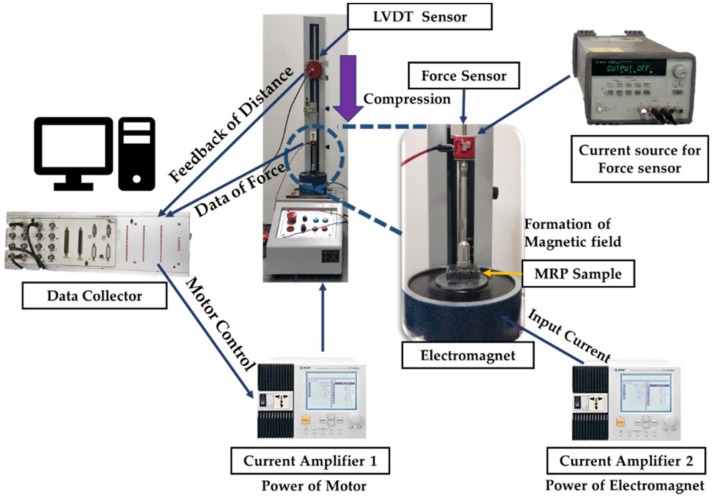
Experimental apparatus of repulsive force measurement with different magnetic fields.

**Figure 9 materials-13-01062-f009:**
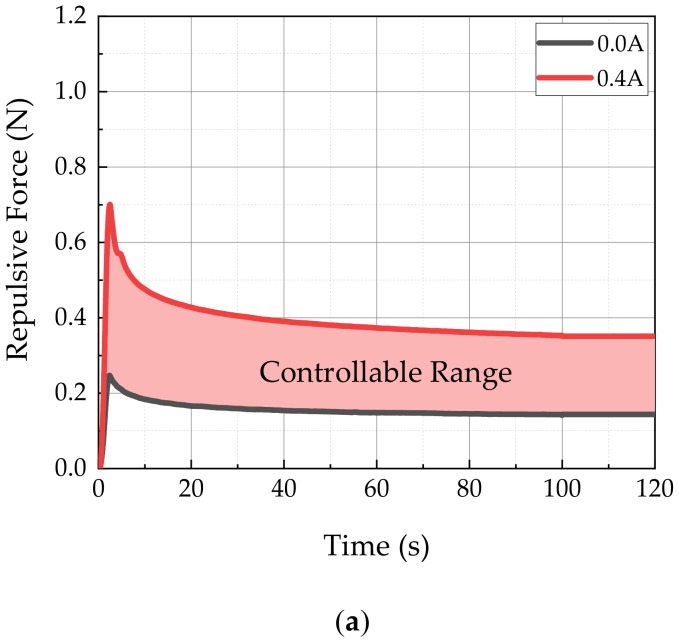
Controllable range: (**a**) MRP1 (25 ppi), (**b**) MRP2 (100 ppi).

**Figure 10 materials-13-01062-f010:**
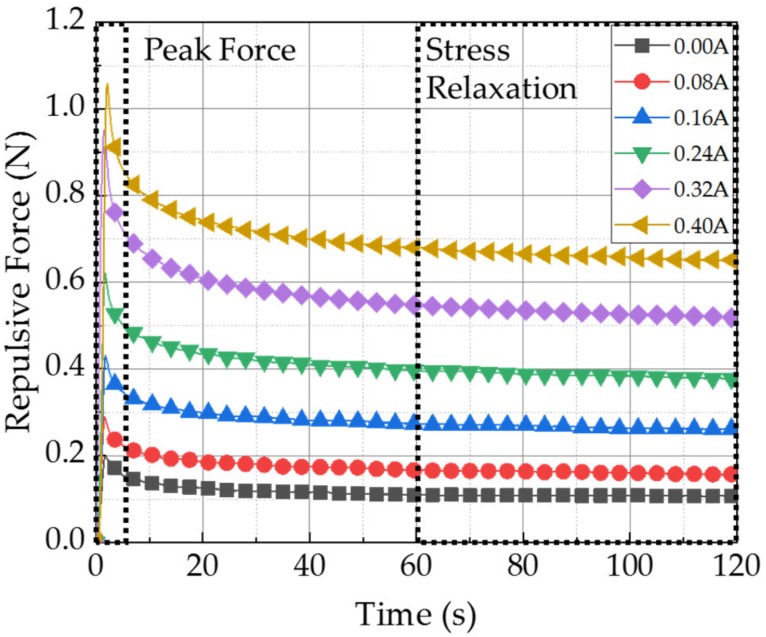
Stress relaxation test results of MRP2 device (1 mm stroke).

**Figure 11 materials-13-01062-f011:**
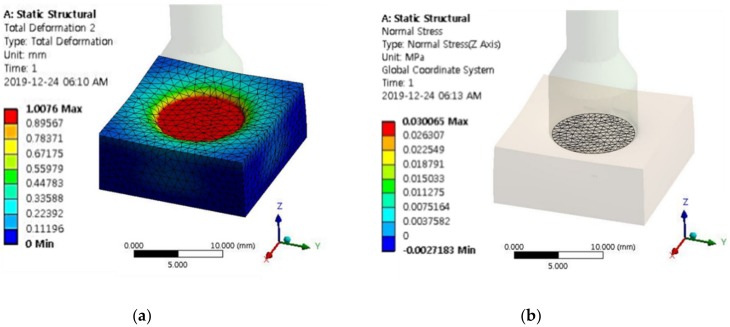
Finite element model to calculate the repulsive force: (**a**) Deformation, (**b**) mesh generation.

**Figure 12 materials-13-01062-f012:**
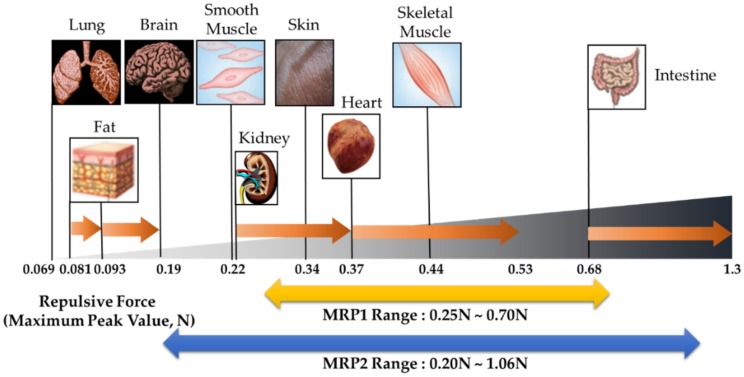
Force spectrum of MRP1 and MRP2 tactile devices with different human tissues.

**Figure 13 materials-13-01062-f013:**
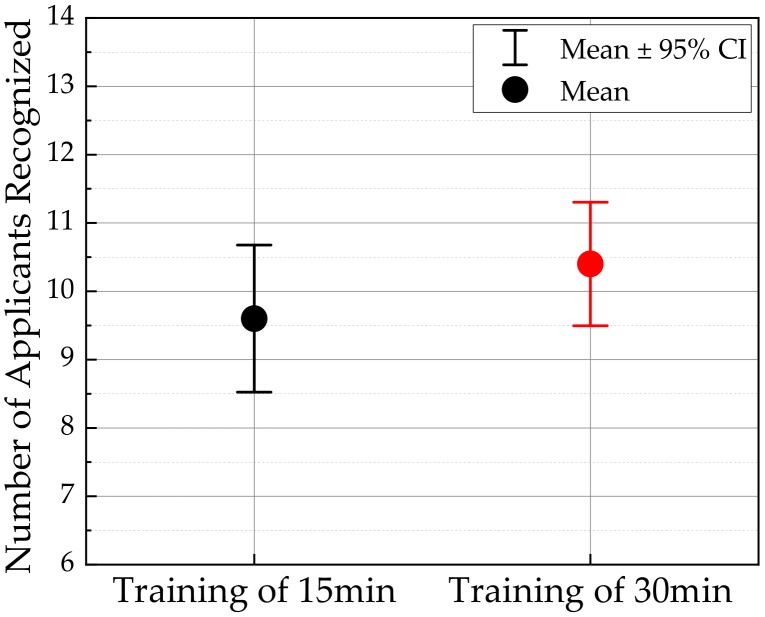
Number of times applicants’ recognition with different training times.

**Table 1 materials-13-01062-t001:** Stress relaxation test results of MRP2 sample (100 ppi, 1 mm, 120 s).

	Input Current	MRP2		Input Current	MRP2
**Maximum Peak Force**	**0.00 A**	0.20 N	**Stress Relaxation**	**0.00 A**	0.11 N
	**0.08 A**	0.29 N		**0.08 A**	0.16 N
	**0.16 A**	0.43 N		**0.16 A**	0.26 N
	**0.24 A**	0.62 N		**0.24 A**	0.38 N
	**0.32 A**	0.95 N		**0.32 A**	0.52 N
	**0.40 A**	1.06 N		**0.40 A**	0.65 N

**Table 2 materials-13-01062-t002:** Data of various human tissues.

	Young’s Modulus of Human Tissues (Pa)	Calculated Maximum Repulsive Force (N)	Magnetic Flux Density Applied to MRP2 (T)
Lung	200	0.069	Not defined
Fat	500–1000	0.081–0.093	Not defined
Brain	1000–4000	0.097–0.19	Not defined
Smooth Muscle	5000	0.22	0.0089
Kidney	5000–10,000	0.22–0.37	0.0089–0.063
Skin	9000	0.34	0.054
Skeleton Muscle	12,000	0.44	0.082
Heart	10,000–15,000	0.37–0.53	0.063–0.10
Intestine	20,000–40,000	0.68–1.3	0.13–0.2 (1.07N–1.3N is not defined)

**Table 3 materials-13-01062-t003:** Psychophysical test survey of MRP2.

Question	No. of Volunteers	Mean	Standard Deviation
Q.1 Can you recognize the difference between repulsive forces?	20	4.25	0.85
Q.2. Do you agree that the repulsive force of each organ can be realized using the proposed sample?	20	4.10	0.91

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
