# Peer review of "A Tactile Device Generating Repulsive Forces of Various Human Tissues Fabricated from Magnetic-Responsive Fluid in Porous Polyurethane"

_materials, 2020, doi:10.3390/ma13051062_

Round 1
Reviewer 1 Report
The article presents the fabrication and testing of a controllable tactile device that can capture varying repulsive forces from human tissues and can be integrated in a robotic surgery system. The device is based on a magnetorheological fluid inserted in a porous polyurethane matrix. In the presence of a magnetic field, the magnetic particles within the fluid align and the fluid changes from Newtonian to a Bingham fluid with a field-dependent yield stress. This coupled with a force sensor serve to measure the response of biological tissues. The article is written in a very comprehensive way. I think that there is merit in the idea that they are proposing and they discuss some of their limitations and ideas to overcome them.
The main objection for this paper is about section 3.4. It is not clear at all what was done, what the training consisted of and whether real samples were being measured or not. Also, the authors should clearly state the goal of this part of the experiment. Is it to prove that a surgeon would be able to use the machine? And if so, if the device gives force magnitude why would a person need to recognize differences?
Also, the article would improve significantly if they measured several biological tissues (even if they are not human) and compared the obtained values to the ones obtained from classical rheological measurements.
Other comments:
The abstract is not very clear Comment on the use of magnetic field inside the human body Comment on the size of the device prototype compared to the size requirements for laparoscopic surgery. Figure 3: include dimensions of the device Figure 4: what is the magnitude of the field? Table 1: How do you define the peak and stress relaxation values? Figure 11: improve resolution Table 2 and 3 should be merged into one.Author Response
Please refer to the attached file

Reviewer 2 Report
Paper can be accepted after the following corrections:
- Figure 2 is not suitable for scientific paper. Please re-draw accordingly to technical standards.
- Figure 6 is not clear. Please indicate magnetic field around the device. Please also make it coherent with the figure 2.
- Figure 8 should be clearly presented. Please present separately the device as well as the schematic block diagram of the system.
Reviewer 3 Report
The manuscript is well written, and it can be accepted for publication after minor revision.
1. Line 100, CHANGE: It is better to say: 2.5 × 2.5 × 1 cm.
2. Line 153, CHANGE: 10 to 50 “um”, used “µm”.
3. The scalse bars of figures 4 and 5 in the manuscript are too small. Please add larger new scale bars in the figures.
4. Line 178, DELETE: x25, and used: SEM images of polyurethane foam: (a) 25 ppi, (b) 100 ppi.
5. Please add suitable scale bars in the figure 7.
Round 2
Reviewer 2 Report
Paper was corrected and can be accepted in the present form.